# Signal-processing and adaptive prototissue formation in metabolic DNA protocells

Avik Samanta [1✉], Maximilian Hörner [2], Wei Liu[1], Wilfried Weber[2] & Andreas Walther [1,3✉]

The fundamental life-defining processes in living cells, such as replication, division, adaptation, and tissue formation, occur via intertwined metabolic reaction networks that process signals for downstream effects with high precision in a confined, crowded environment. Hence, it is crucial to understand and reenact some of these functions in wholly synthetic cell-like entities (protocells) to envision designing soft materials with life-like traits. Herein, we report on all-DNA protocells composed of a liquid DNA interior and a hydrogel-like shell, harboring a catalytically active DNAzyme, that converts DNA signals into functional metabolites that lead to downstream adaptation processes via site-selective strand displacement reactions. The downstream processes include intra-protocellular phenotype-like changes, prototissue formation via multivalent interactions, and chemical messenger communication between active sender and dormant receiver cell populations for sorted heteroprototissue formation. The approach integrates several tools of DNA-nanoscience in a synchronized way to mimic life-like behavior in artificial systems for future interactive materials.

[1] A3BMS Lab, University of Mainz, Department of Chemistry, Duesbergweg 10-14, 55128 Mainz, Germany. [2] Faculty of Biology, Cluster of Excellence CIBSS - Centre for Integrative Biological Signalling Studies, University of Freiburg, 79104 Freiburg, Germany. [3] Cluster of Excellence livMatS @ FIT – Freiburg Center for Interactive Materials and Bioinspired Technologies, University of Freiburg, 79110 Freiburg, Germany. ✉email: avik.samanta@uni-mainz.de; andreas.walther@uni-mainz.de

Living cells execute numerous complex functions, such as division, differentiation, adaptation, and tissue formation, by guiding the flux of materials and information via intertwined metabolic reaction networks in a crowded intracellular environment or even in liquid/liquid phase-segregated (LLPS) membraneless organelles[1–5]. In recent years, the bottom-up reenactment of fundamental cellular processes inside artificial synthetic compartments has emerged as an appealing strategy to investigate biological pathways from molecular to macroscopic length-scale in a controlled environment[6–9]. These approaches not only pave the way towards constructing a wholly man-made synthetic cellular equivalent (protocells, PCs) with life-like functionality and adaptivity but also provide an insight into fundamental reaction network-driven processes in natural cells. Historically, liposomes[10–15], polymersomes[16–18], colloidosomes[19,20], and proteinosomes[21] based PC models have been used for encapsulation of bio-inspired transformations or also communication networks. This involves, for instance, DNA-mediated signal cascades, self-replication, and in-vitro protein synthesis[22]. However, such systems may suffer from low and heterogeneous encapsulation efficiency, absence of molecularly crowded interior resembling the cytoplasmic matrix, and lack control over stoichiometry of components in the case of multimeric active cargoes. In contrast, PC models[23–26] based upon complex coacervates, aqueous two-phase systems, and LLPS of intrinsically disordered proteins have been suggested as more appropriate cytoplasmic model systems because their inherent macromolecular-rich crowded interior and high loading capacity resemble the cytoplasmic matrix[27–33]. Various studies reported the use of such macromolecularly crowded PCs for gene expression[34], ribonucleic acid catalysis[35,36], and multienzyme iterative processing in multicomponent microdroplets[37,38]. In spite of these recent examples, functional adaptation in crowded PCs using metabolic transformations based upon an external signal using an encapsulated catalyst with precisely organized downstream action, such as "phenotype-like changes" or communication and prototissue[39,40] formation, remain challenging and are rarely explored[19]. Recently, we reported the functional and morphological adaptation ability of all-DNA PCs[41] via in situ generation of a self-reporting non-DNA metabolite upon ring-closing metathesis reaction based on genetically modified artificial metalloenzymes immobilized in PCs[42]. The non-DNA metabolite could interact with double-stranded DNA (dsDNA) in the shell, which led to metabolic growth, mechanical stress built up, and ultimately PC fusion. Since adaptivity is imperative for life's survival in a dynamic environment, it is of critical importance to establish strategies capable of converting chemical signals from diverse origins to allow for intra- or inter-PC downstream processes such as functional adaptation and communication.

DNA has become a relevant biomacromolecule for nanoscience and systems chemistry research for its programmable structure formation and its ability to rationally design reaction networks and concatenated logic operations, demonstrating its potential as a toolbox in molecular computing[43–49]. Among other DNA principles, catalytic nucleic acids (DNAzymes and ribozymes)[50–53] have emerged as one relevant tool in the DNA reaction network toolbox. Towards the first combination of DNAzymes with PCs, recently, it has been shown that the ribozyme activity of bond cleavage is enhanced inside membraneless coacervate droplets[36]. However, a higher-level function integration of such systems in protocell research remains elusive.

Herein we introduce catalytic signal processing inside highly programmable all-DNA protocells (PC) formed by LLPS of specific single-stranded DNA (ssDNA) strands. As a key tool, we encapsulate a DNAzyme that cleaves RNA positions in DNA-RNA chimera substrates that serve as signals. We demonstrate enhanced metabolic signal conversion by RNA bond cleavage inside the PCs and capitalize on this understanding by using

the signal output to instigate downstream phenotype-like changes in the PC, as well as prototissue formation by PC clustering via induction of multivalent self-complementary interaction at the PC surface. Moreover, we also show an interprotocellular communication of the metabolites between catalytically active sender PCs and inactive receiver PCs, leading to the formation of largely sorted prototissue-like colonies. All strategies are enabled by engaging the metabolized output of the initial signal downstream in dynamic DNA strand displacement (DSD) reactions.

## Results and discussion

Building upon our previous work[41,42,54], we prepared PCs composed of liquid ssDNA interiors and DNA hydrogel shells by rapid self-compartmentalization during a temperature ramp (ca. 10 min) of an aqueous solution containing two ssDNA-multiblock copolymers (p($A_{20}$-m) and p($T_{20}$-n)). $A_{20}$ and $T_{20}$ denote homo-repeats of 20 adenine and 20 thymine nucleotides (nt), while m and n stand for defined, so-called barcode domains, which are used for functionalization using their complementary counterparts (Fig. 1a). In short, during heating, the $A_{20}/T_{20}$ duplexes dissociate, and p($A_{20}$-m) undergoes LLPS during the heating step, while p($T_{20}$-n) remains dissolved. During cooling, re-hybridization of $A_{20}/T_{20}$ happens rapidly at the periphery of the p($A_{20}$-m) phase-separated droplets, giving rise to a hydrogel-like shell stabilized by $A_{20}/T_{20}$ duplexes. Eventually, the p($A_{20}$-m) dissolves below its cloud point temperature (ca. 45 °C)[41], but stays entrapped in a liquid state under high osmotic pressure and macromolecular crowding, as confined by a compact hydrogel-like shell. Apart from the facile preparation and providing a cytosol-like DNA-based crowded, confined platform, one of the significant advantages of such PCs is the selective addressability and orthogonal functionalization of the interior and the shell using the barcode domains (m,n) (Supplementary Fig. 2 and Supplementary Table 1).

To realize the catalytic metabolism inside the PCs, we introduced a specific $Mg^{2+}$-dependent RNA-cleaving DNAzyme (Dz). This specific DNAzyme cleaves a single RNA linkage embedded in a DNA substrate efficiently[52]. For the facile loading of Dz in the PC core, we separated it into two ssDNA sequences, so-called split DNAzymes (Figs. 1a, 2a and Supplementary Fig. 1). Each sequence contains one-half of the complementary barcode domain ($m_{1/2}$*), a stem part (s or s*), an active site sequence ($t_1$ or $t_2$), and a substrate docking site ($u_1$ or $u_2$). After annealing both parts to form the acting Dz, these Dzs were added to a buffered PC dispersion containing 50 mM $Mg^{2+}$ (for all the experiments hereafter), whereupon the Dzs rapidly dock inside the PC core via binding with every m-barcode sequence present in the p($A_{20}$-m) chains. Since the concentration of m-barcode in the PC interior can be precisely determined, all the barcodes can be easily functionalized with Dzs using proper amounts added to the dispersion. Any excess Dz in the solution is removed by washing the PCs via centrifugation.

Figure 1 summarizes the analysis of the catalytic activity of the Dz-loaded PCs (Dz⊂PCs), as well as the strategies for downstream signal processing in a step-by-step manner using three different thoughtfully designed substrates (Fig. 1b): (1) A self-reporting substrate (Subs-1) with an RNA linkage, in which a fluorescence resonance energy transfer (FRET) pair (fluorophore + quencher) is separated upon bond cleavage helps to analyze the catalytic conversion. (2) Subs-3 with a cleavage domain at its loop (Supplementary Fig. 3), concealing a sequence (n*) complementary to the shell barcode that is released upon catalytic loop cleavage, is used for the phenotype-like change by DSD of shell-immobilized shorter Atto$_{488}$-$n_{short}$*. (3) A loop substrate Subs-4 with a palindromic sequence connected to n*

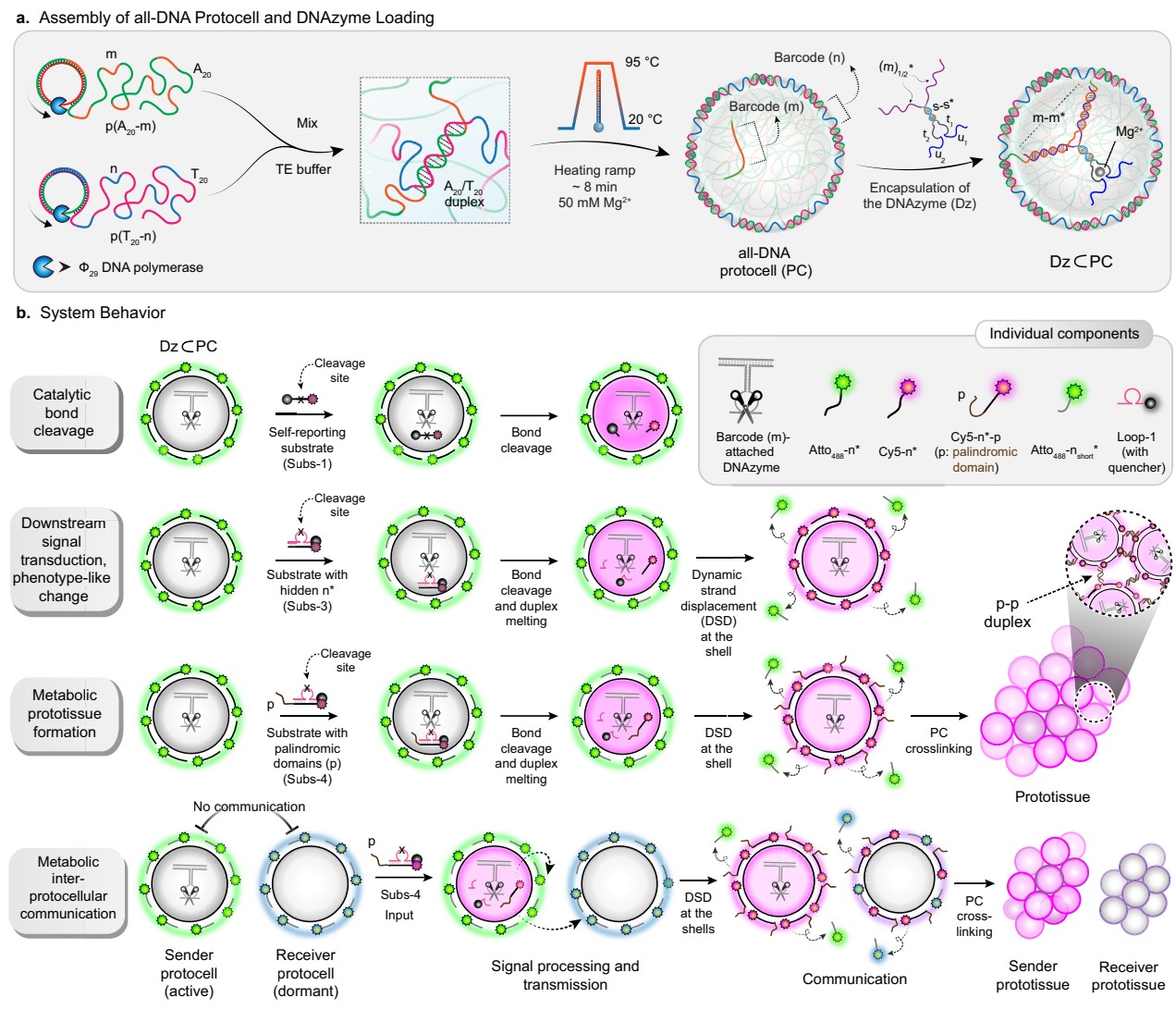

**Fig. 1 Design, strategy, and system behavior of DNAzyme-catalyzed signal conversation, downstream functional adaptation inside all-DNA PCs, metabolic prototissue formation, and interprotocellular communication. a** Synthesis of sequence-specific two multiblock ssDNA polymers via rolling circle amplification (RCA). A buffered mixture of the ssDNA polymers is subjected to a fast-heating ramp (3 °C/min) in the presence of $Mg^{2+}$ (50 mM). Heat-induced phase-separation of $p(A_{20}-m)$ during heating and antagonistic duplex ($A_{20}-T_{20}$) hybridization (at the coacervate surface) during the cooling step, resulting in the self-assembly of kinetically-trapped all-DNA PCs. The core and shell barcodes are denoted with m and n, respectively. The immobilization of DNAzymes (Dz) is achieved using duplex hybridization between m* residues of the Dz and the core barcodes (m). **b** The system properties are demonstrated in three different pathways. Firstly, a self-reporting substrate (Subs-1) carrying an RNA linkage is used to investigate the catalytic efficiency of Dz in the confined state, in which Cy5 fluorescence (magenta color) increases upon cleaving the RNA linkage. Secondly, the downstream signal processing and phenotype-like change in PCs are demonstrated using a caged substrate (Subs-3). The Dz-catalyzed bond cleavage at the Loop-1 and subsequent release of Cy5-n* substitutes $Atto_{488}-n_{short}*$, changing the shell color from green to magenta. Thirdly, the metabolic prototissue formation is demonstrated by a similar loop substrate (Subs-4) bearing a palindromic (p) sequence, which gets exposed at the PC periphery after the intra-protocellular metabolic bond cleavage and a DSD reaction at the shell, leading to crosslinking of the PCs via p–p duplex formation. Lastly, the formation of sender and receiver prototissue formation via downstream interprotocellular signal transduction from active to dormant PCs is prompted by catalytic bond cleavage of Subs-4. All the strands are listed in Supplementary Table 2.

(Cy5-n*-p) is used to prompt inter-PC communication by multivalent binding scenarios.

Firstly, to investigate the catalytic activity of the Dz inside and outside the PCs, we used a self-reporting substrate (Subs-1) with an RNA linkage and a FRET pair (Cy5 and a black hole quencher (BHQ)) covalently attached at both ends. Upon Dz-catalyzed breaking of the RNA bond in Subs-1, Cy5, and BHQ are separated (Fig. 2a), giving rise to the enhancement of fluorescence intensity (represented in magenta color). Spectrofluorometric (for investigating catalytic efficiency), cytometric (for statistical analysis), and microscopic (for real-time visualization) techniques

were used purposefully to characterize the intra-protocellular catalysis and to optimize the sequences for the next step. The stoichiometric ratio of Subs-1 to Dz is fixed at 2.5 for comparing the catalyst activity. In more detail, Subs-1 binds to the $u_1$ and $u_2$ residues of the Dz, but after cleavage, the two parts of the initial ssDNA signal melt away due to the low melting temperature ($T_m = \sim 20$ °C) of each individual half. The corresponding fluorescence traces show a 2-fold more efficient Dz-catalyzed bond cleavage (DCBC) in the PC (~62% turnover of Subs-1) as compared to the solution (~35% turnover of Subs-1). The higher activity of the DNAzyme inside the PC can be addressed by the

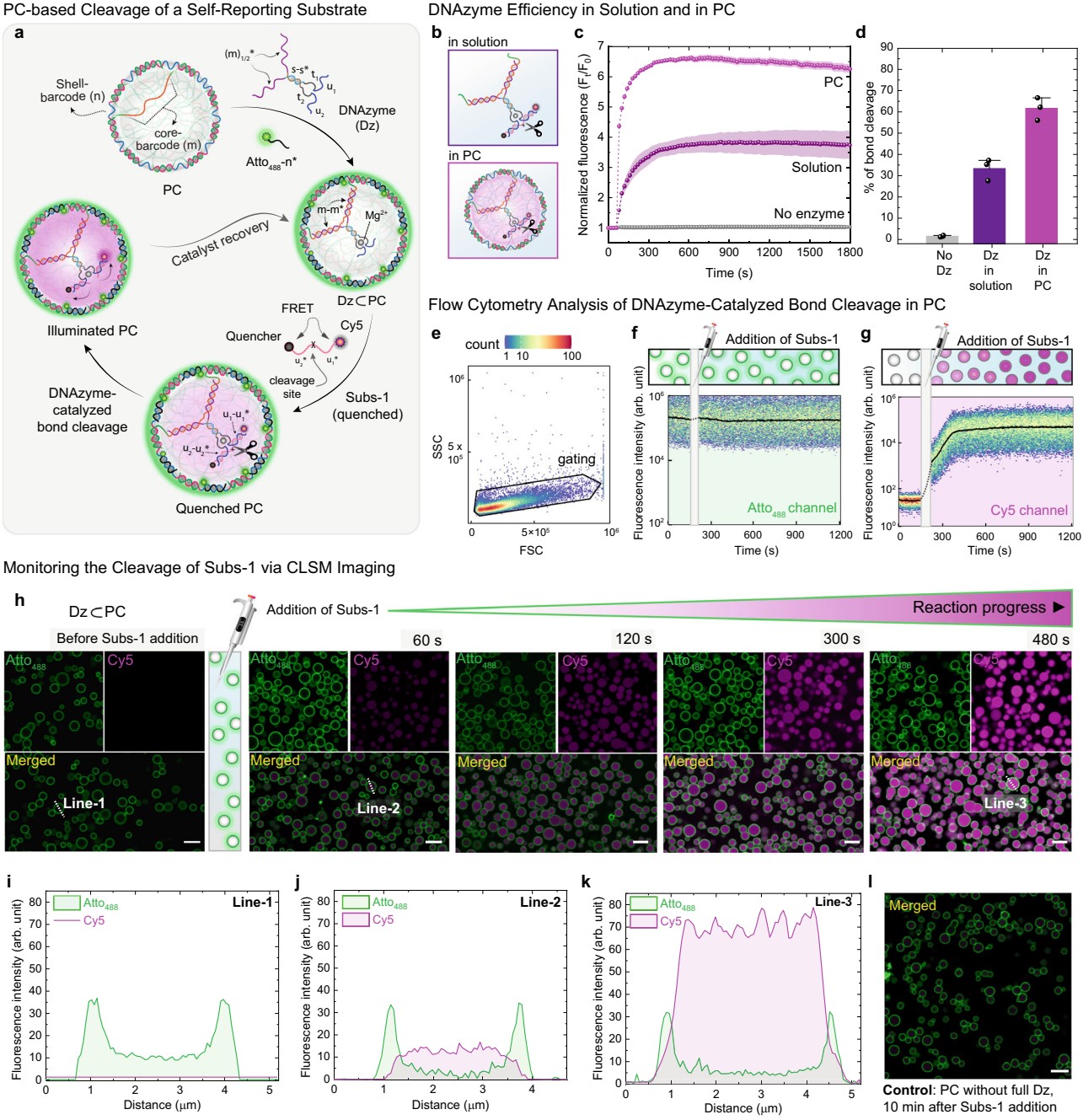

**Fig. 2 Intra-protocellular catalytic cleavage of self-reporting substrate. a** Schematic representation of a catalytic cycle for RNA linkage cleavage in Subs-1 in PC interior and fluorescence enhancement upon uncaging the Cy5-appended product. **b** Schematic representation of two scenarios, where the DCBC is performed in (i) a buffered solution where the Dz is hybridized with a monomeric m-sequence, and (ii) in DzꞇPCs. **c** Time-dependent spectrofluorimetric investigation of DCBC in PC and solution. A control experiment in which the auto-cleavage of Subs-1 is also presented in the absence of Dz (gray trace). **d** The percentage of bond cleavage of Subs-1 in the three scenarios mentioned above. **e** PCs were identified in the forward scatter (FSC) versus side scatter (SSC) plot. **f**, **g** Intra-protocellular Atto$_{488}$ and Cy5 fluorescence were monitored over time by flow cytometry. Black line shows the moving median. **h** The time-dependent CLSM images for DCBC induced Cy5-appended sequence uncaging in an Atto$_{488}$-labeled PC system. The shell of all the PCs is labeled with Atto$_{488}$-n* (green channel). At $t = 0$ (before the addition of Subs-1), all the PCs are visualized as green circles since the fluorescent Atto$_{488}$-n* is hybridized with the shell barcode domain ($n$) of p(T$_{20}$-n) polymer. The DCBC in DzꞇPCs is monitored over 8 min time period, and a gradual increase of Cy5 fluorescence (magenta channel) at the PC core is observed. **i–k** Line segment analysis in both channels (green and magenta) of Line-1, Line-2, and Line-3 at $t = 0$, 1, and 8 min, respectively. **l** CLSM image of dormant PCs (without the Dz, only with Split-Dz-1) at $t = 10$ min after adding Subs-1. Data were the means ± standard deviation of duplicate reactions. Scale bars: 5 µm. The error regions in panel **c** represent the standard deviation of two independent experiments. The error bar in **d** represents the standard deviation of three independent experiments ($N = 3$). Condition: for spectroscopic measurements: [DzꞇPCs], [Dz in solution] $= 2$ µM, [Subs-1] $= 5$ µM, for cytometry experiments: [DzꞇPCs] $= 0.5$ µM, [Subs-1] $= 1.3$ µM and for the microscopic experiments: [DzꞇPCs] $= 7$ µM, [Subs-1] $= 16.6$ µM, at 25 °C, TE buffer.

crowding effect of the liquid DNA core and the different environmental conditions that the DNAzyme experiences inside the PC compared to those in the solution. The polarity of the liquid DNA environment and high ionic strength could play an important role. The reaction rates are also faster (Fig. 2c), and the turnover number is ca. 1.6, indicating multiple turnovers per Dz. We, however, also observe that the encapsulated Dz⊂PCs reduce its activity beyond this turnover, which indicates that the cleaved product reaches an equilibrium between the free and bound state due to its stickiness to the catalytic site at 25 °C (Supplementary Fig. 4). The slight decrease in the fluorescence intensity of the cleaved product after 15 min of substrate addition can be attributed to the PC sedimentation at the bottom of the well plate (Fig. 2c). The negative control shows almost no product formation in the absence of the Dz.

To achieve more detailed insights into the level of the PC population, we performed flow cytometry experiments. In this case, the shells of Dz⊂PCs (p(T$_{20}$-n)) were labeled with Atto$_{488}$-n* (green color in Fig. 2a, f), and Subs-1 was added after ~2 min of measurement. PCs were identified in the forward scatter (FSC, size) versus side scatter (SSC, complexity/granularity) plot, and signals without Atto$_{488}$ and Cy5 fluorescence were excluded from the analysis (<3% of all protocells, Fig. 2e). The fluorescence intensity in the Cy5 channel increases after the addition of Subs-1 and reaches a plateau after 350 s (Fig. 2g). This corresponds to the Dz-catalyzed cleavage of Subs-1. In contrast, the Atto$_{488}$ fluorescence of the shell remains unaltered over the measurement window (Fig. 2f). The consistent evolution of the cytometry data underscores a homogeneous PC population. Control experiments with dormant PCs (without the whole Dz) do not show a steady increase in fluorescence in the Cy5 channel (Supplementary Fig. 5).

Furthermore, we selected the Atto$_{488}$-labeled Dz⊂PCs for in situ morphological studies using confocal laser scanning microscopy (CLSM). The PCs are initially visualized with a green shell and empty core ($t = 0$ min, Fig. 2h). Once Subs-1 is injected into the PC suspension, a magenta fluorescence (excitation 637 nm) appears in the core of the Dz⊂PCs, which confirms that the compartmentalized DNAzyme catalyzes the RNA bond cleavage, producing the Cy5-appended output. The rate of product formation correlates with both fluorescence spectroscopy and the cytometry results (Fig. 2c, g). Interestingly, the magenta fluorescence persists in the PC core after the completion of the reaction, and an immediate diffusive leveling into the solution does not occur at 25 °C, which is also supported by the cytometry results (Fig. 2g). The line segment analysis of the confocal micrographs before the substrate addition (0 min, line-1), after the addition (1 min, Line-2), and at the reaction completion (8 min, Line-3) exhibit an increase of magenta fluorescence along the lines throughout the reaction (Fig. 2i–k). In contrast, the green fluorescence at the shell remains unchanged. In a relevant control experiment, no Cy5 fluorescence enhancement is observed in the PC core when the same amount of Subs-1 was injected into a dormant PC (without the encapsulated Dz; Fig. 2l).

To better understand the diffusivity of the output product of the DCBC and the potential binding to the substrate recognition site (u1 and u2), we performed FRAP (fluorescence recovery after photobleaching) cycles inside the Dz⊂PC interior. The repeated photobleaching on a PC after completion of the reaction (cleaving the Subs-1) in both Atto$_{488}$ (green) and Cy5 (magenta) channels exhibits different diffusion patterns. The green fluorescence at the PC shell (region of interest 1, ROI-1) does not show any recovery after every bleach cycle and attains a fully bleached state after the fifth cycle (Fig. 3a, b). This corresponds to a hydrogel-like shell structure with negligible fluorophore diffusion. In contrast, the magenta fluorescence at the PC core (ROI-3) exhibits almost complete recovery after each bleach cycle, confirming rapid

diffusion of the product inside the PC (Fig. 3a, c). Note that a control area outside of the PC (ROI-4) also shows bleaching of Cy5 fluorescence and recovery due to some slowly leaked product (Supplementary Fig. 6).

Furthermore, to demonstrate the continued operation and also refreshed use of the encapsulated Dz, we injected a second substrate (Subs-2) with a different fluorophore (Cy3) to the Dz⊂PCs after completion of the Subs-1 cleavage (Fig. 3d). Prior to the addition of Subs-2, the release of the cleaved product Subs-1 (with Cy5) from the Dz active site and the PC core is confirmed using CLSM images (Fig. 3e, f). A detailed 1-h kinetic study on the product release from the PC is shown in Supplementary Fig. 7. The line segment analysis obtained from CLSM reveals 82% of product releases from the PCs in 1 h (Fig. 3g, h). Upon addition of Subs-2, the Cy3 fluorescence (cyan) increases rapidly inside the PC core, indicating the cleavage of the RNA linkage in Subs-2 (Fig. 3i). Figure 3j presents the line segment analysis of the three channels (Atto$_{488}$, Cy3, and Cy5) after 8 min of the Subs-2 addition. These observations corroborate that the Dz, immobilized in the PC core, can be used for multiple catalytic cycles of bond cleavage, which is crucial for a metabolic system.

After investigating the catalytic activity of the Dz and the product diffusion in a compartmentalized state, we set out to establish an intra-protocellular downstream adaptation pathway, in which a signal, released via metabolic bond cleavage, triggers a cascade leading to a functional phenotype-like change in the PCs. In this regard, we used a loop-containing substrate (Subs-3), in which a Cy5-n* sequence (black)— complementary to the shell barcode n of the PC—is concealed using another ssDNA (pink) carrying the RNA linkage at its Loop-1 domain and a BHQ at the 3′ end (Fig. 4a, b). The Cy5 fluorescence is quenched in Subs-3 because the Cy5 of n* and the quencher of Loop-1 are in proximity in the hybridized state (Supplementary Figs. S1 and S3). Two mismatches were strategically placed at the duplex domains on either side of the loop to keep the $T_m$ around 39 °C so that both parts of cleaved Loop-1 melt away as soon as the loop is cleaved by the Dz. The loop of Subs-3 consists of 11 nt with one RNA linkage at the middle and is designed to be recognized by the Dz active site. The time-dependent Cy5 fluorescence upon cleavage of the RNA linkage at the loop domain of Subs-3 was monitored via fluorescence spectroscopy after injecting a solution of Subs-3 in three separate scenarios: (i) Dz encapsulated inside the PC (Dz⊂PCs), (ii) Dz are in solution hybridized with monomeric barcode, and (iii) in PCs in a buffered solution without the Dz (Fig. 4b, c). Upon addition of Subs-3, the Cy5 fluorescence increases for scenarios (i) and (ii), indicating the DCBC and subsequent bond cleavage driven duplex melting (BCDM). It is evident that the metabolic release of Cy5-n* is 1.5 times higher in the case of Dz⊂PCs, and occurs at a faster rate than the reaction in solution. This observation corroborates the fact that the catalytic efficiency is higher inside the PCs. The fluorescence intensity increases negligibly in the case of the negative control (no Dz; (iii)), indicating less than 5% of the thermal leakage of Subs-3 at the experimental temperature (37 °C).

To demonstrate the metabolic signal transduction in the PCs, we combined the two consecutive processes, DCBC and BCDM, to release Cy5-n* from Subs-3 inside the Dz⊂PCs, which subsequently diffuses towards the shell and triggers a downstream DSD reaction. The DSD reaction leads to a substitution of the Atto$_{488}$-n$_{short}$* that was initially present to label the shell in green. This DSD induces a color change in the PC shell due to tighter anchoring of the Cy5-n* (Fig. 4a). This corresponds to a change in the phenotype of the PC. The DSD is enabled using an Atto$_{488}$-n$_{short}$*, which only features a complementarity of 13 nt to the n barcode domain ($n = 21$ nt), leaving a toehold domain of 8 nt free for the DSD. Flow cytometry

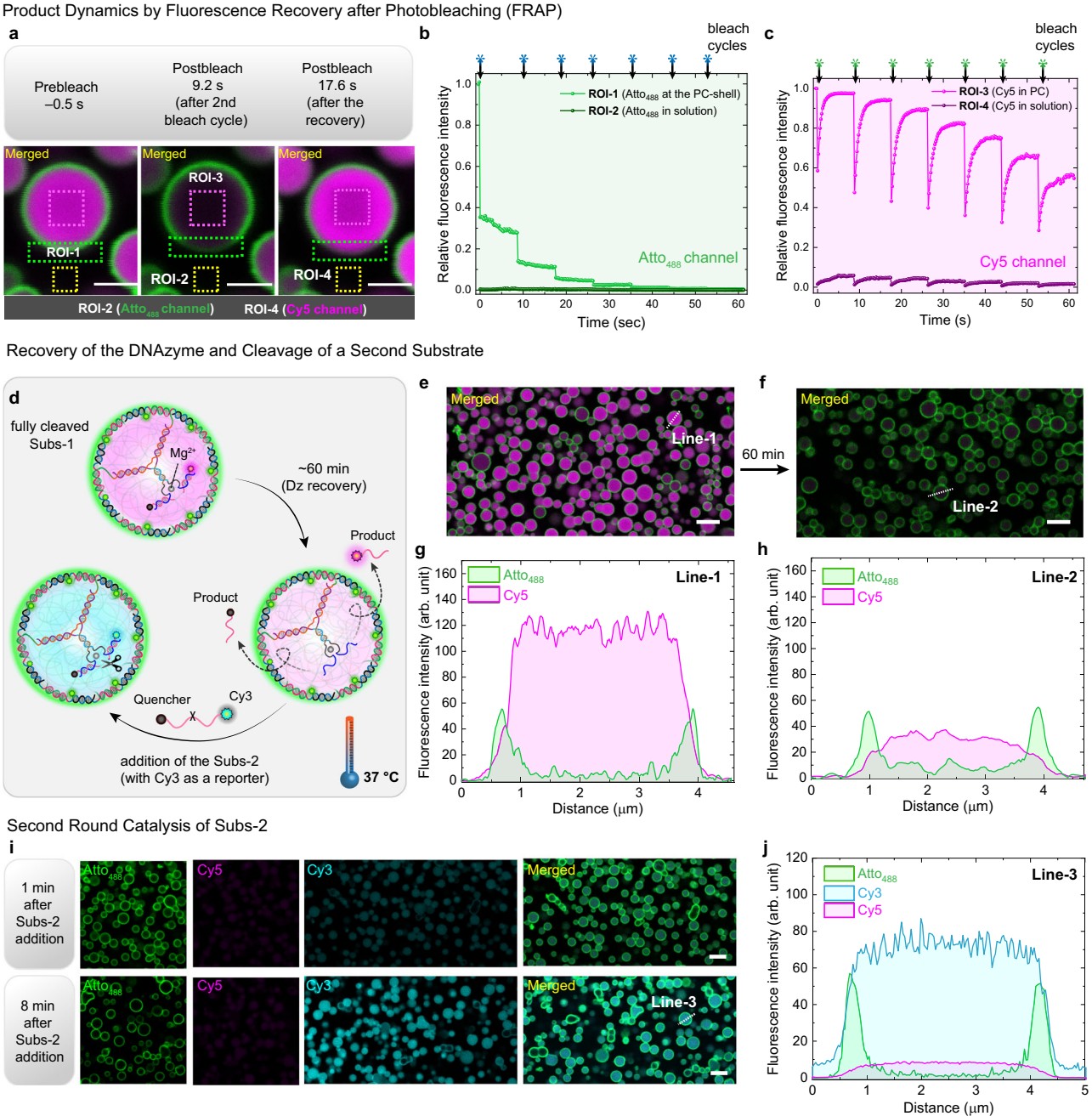

**Fig. 3 CLSM investigation of intra-protocellular product diffusion and catalyst recovery. a** CLSM image of a Dz⊂PC after the completion of DCBC of Subs-1 and the photobleaching is performed within the dotted rectangles. The row of CLSM images depicts the pre-bleach state, the post-bleach state immediately after the second bleach cycle, and another post-bleach state after the recovery, hence just before the third bleaching cycle. **b** The relative fluorescence intensities from ROI-1 and ROI-2 (green channel for Atto$_{488}$) over the seven bleach cycles. **c** The relative fluorescence intensities from ROI-3 and ROI-4 (magenta channel) over seven bleach cycles, depicting Cy5 fluorescence recovery. The black downward arrows represent the bleaching events. Scale bars: 3 μm. **d** Schematic representation of catalyst recovery after completion of the Subs-1 cleavage and the confirmation of the free DNAzyme active sites using Subs-2, a self-reporting substrate with a second fluorophore (Cy3, cyan). **e** CLSM image of Dz⊂PCs with fully-cleaved Subs-1 (merged channel: Atto$_{488}$ and Cy5). **f** CLSM image of Dz⊂PCs after 60 min of Subs-1 addition. **g, h** The line segment analysis in both channels of Line-1 and Line-2, before and after the product release from PC core, respectively. **i** CLSM images Dz⊂PCs after the addition of Subs-2 ($t = 1$ min and $t = 8$ min). Three channels represent the PC shell (green, Atto$_{488}$), the residual product of Subs-1 (magenta, Cy5), and the product generated from Subs-2 (cyan, Cy3). Scale bars: 5 μm. **j** The line segment analysis of Line-3 of three channels. Conditions: [Dz⊂PCs] = 5 μM, [Subs-1] and [Subs-2] = 12.5 μM in TE buffer at pH 8.

analysis of Atto$_{488}$-n$_{short}$*-labeled Dz⊂PC shows a rapid increase in Cy5 fluorescence (magenta channel) upon injection of Subs-3 (Fig. 4e), whereas the fluorescence of Atto$_{488}$ (green channel) gradually decreases over time (Fig. 4d). These observations indicate that (i) the Cy5-n* sequence is catalytically released from the input signal (Subs-3) upon DCBC and BCDM in the PC core, leading to an

enhancement of the magenta fluorescence, and (ii) that the DSD reaction at the PC shell occurs via ejection of the Atto$_{488}$-n$_{short}$* from the PC shell, resulting in a decrease of the green fluorescence. It is essential to realize that the product is now removed from any substrate/product binding competition at the Dz in the PC interior by spatial relocation into the shell. In a control experiment, when a

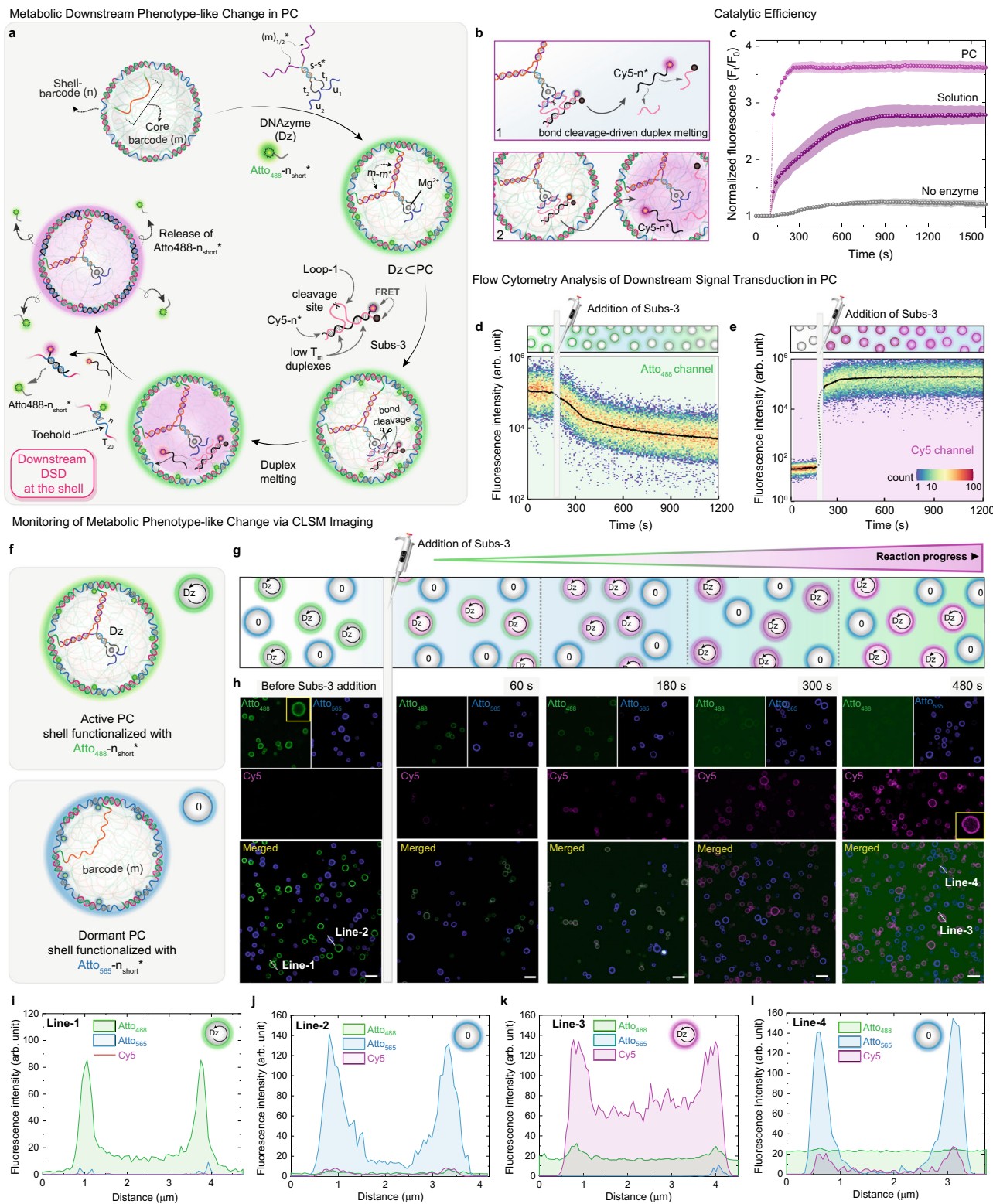

Subs-3 solution is added to a dormant PC flow (without the whole Dz), no change in fluorescence at the Atto$_{488}$ channel (green) is observed (Supplementary Fig. 8).

Next, we investigated the Dz-catalyzed downstream signal processing in Dz⊂PCs using in situ CLSM (Fig. 4f–h and Supplementary Fig. 9). To clearly monitor changes, we mixed active PCs (Atto$_{488}$-n$_{short}$*-labeled Dz⊂PCs; shell in green) with dormant Atto$_{565}$-n$_{short}$*-labeled PCs (without the Dz; shell labeled in blue; Fig. 4f, g). Upon adding Subs-3 into the mixed-PC

dispersion, the Cy5 fluorescence (magenta channel) appears exclusively inside the active Dz⊂PC cores and close to the shell, while the dormant magenta PCs remain dark in the core. Within 5 mins of the substrate addition, the color of the active Dz⊂PC shells changes from green to magenta, confirming the downstream DSD substitution of Atto$_{488}$-n$_{short}$* in the shell. The Atto$_{488}$-n$_{short}$* is released into the surroundings and increases the background fluorescence. In contrast, the shell color of the dormant PCs remains unaltered after 8 min, indicating the absence of

**Fig. 4 Intra-protocellular DNAzyme-catalyzed bond cleavage and ensuing downstream signal processing, leading to PC phenotype-like change.**
**a** Schematic representation of the Dz-catalyzed RNA linkage cleavage of Subs-3 and subsequent duplex melting, releasing the Cy5-n* strand in situ, which triggers a downstream DSD reaction at the shell of Atto$_{488}$-n$_{short}$*-labeled Dz⊂PCs, changing the shell-fluorescence from green to magenta. **b** Schematic representation of two scenarios, where the DCBC and subsequent BCDM were performed (1) in solution where Dz is hybridized with a monomeric m-sequence and (2) in Dz⊂PCs using Subs-3. **c** The spectrofluorimetric analysis of DCBC and downstream BCDM in the case of Dz in solution (dark purple trace) and at its encapsulated state (magenta trace). The thermal uncaging of Subs-3 (in the absence of Dz) is shown in a gray trace. **d, e** Time-dependent cytometric analysis of intra-protocellular DCBC, BCDM, and DSD reaction via monitoring change in PC fluorescence intensity of Atto$_{488}$ and Cy5, respectively. The gradual decrease of Atto$_{488}$ fluorescence and rapid increase in Cy5 fluorescence indicates the metabolic phenotype-like change in Dz⊂PCs. **f, g** Schematic representation of DCBC, downstream BCDM, and DSD reaction in a mixed-PC system, containing active Atto$_{488}$-n$_{short}$*-labeled (green) Dz⊂PCs and dormant Atto$_{565}$-n$_{short}$*-labeled (blue) PCs. **h** Time-dependent CLSM images of metabolic phenotype-like change in the mixed-PC system. Upon Subs-3 addition, a gradual increase of magenta fluorescence is observed at the shell of the active green PCs substituting the green fluorescence, while the blue dormant PCs remain unchanged. **i, K** The line segment analysis on an active PC before (Line-1) and after (Line-3) the reaction and downstream transformation, respectively. **j, l** The line segment analysis on a dormant PC before (Line-2) and after (Line-4) the reaction and downstream transformation, respectively. Scale bars: 5 μm. The error regions in panel **c** represent the standard deviation of two independent experiments. Condition: for spectroscopic measurements: [Dz⊂PCs], [Dz in solution] = 2 μM, [Subs-3] = 6 μM, for cytometry experiments: [Dz⊂PCs] = 0.5 μM, [Subs-3] = 1.5 μM and for the microscopic experiments: [Dz⊂PCs] = 7 μM, [Subs-3] = 16.6 μM, at 37 °C, TE buffer at 35 °C.

a metabolic reaction network to produce Cy5-n*. The line segment analyses on active Dz⊂PCs before and after the reaction are shown in Fig. 4i, k (Line-1 and Line-3), representing the decrease of green fluorescence (Atto$_{488}$-n$_{short}$*) at the PC shell and enhancement of magenta fluorescence (Cy5) at the core and mainly at the shell of the active PCs. The intensity of the blue shell-fluorescence of the dormant PCs remains very similar before and after the reaction (Fig. 4j, l; Line-2 and Line-4). This stability of the blue fluorescence in the dormant PCs also underscores hardly any diffusive exchange of the Cy5-n* between active and dormant PCs at this substrate concentration. In a control experiment, when a high concentration Subs-3 solution (10 equivalent to the shell barcode) is added to a dormant PC dispersion, no DSD at the PC shell, hence no decreases in the shell-fluorescence, is observed (Supplementary Fig. 10a).

Moreover, we envisaged a more drastic downstream adaptation scenario by installing communication amongst active PCs via the metabolic presentation of an attractive multivalent interaction between them. In this context, we designed a Subs-4, which is similar to Subs-3 but features an additional palindromic sequence (6 nt = CTC GAG) attached to Cy5-n* (Fig. 5a). The $T_m$ of the palindrome (p) is well below room temperature (17 °C), so Subs-4 does not form homodimers at the experimental temperature (~35 °C). However, we hypothesized that p could induce inter-PC attraction by multivalent p–p interactions. Indeed, upon the addition of Subs-4 into a dispersion of Atto$_{488}$-n$_{short}$*-labeled Dz⊂PCs, the shell color first changes over time from green to magenta due to the release of Cy5-n*-p (from the PC core) and ensuing DSD at the shell. Critically, due to the localization of the Cy5-n*-p at the shell, the emerging magenta PCs start forming prototissue-like aggregates via multivalent inter-PC p–p duplexes (Fig. 5b). In a control experiment, when 10 equivalent (to the barcode n concentration) of Subs-4 is added to a dormant PC suspension, where the PCs do not contain any Dz, no substantial PC assembly is observed (Supplementary Fig. 10b). These control experiments confirm that catalysis is a critical aspect of the downstream process.

To obtain a quantitative understanding of the downstream prototissue formation, we set out to correlate the size of the prototissues (that is the number of PCs in a prototissue) with the surface density of the palindromic strand at the PC shell. To this end, we varied the palindrome surface density by co-functionalization of the n barcodes in the p(T$_{20}$-n) shells with the palindromic Cy5-n*-p (magenta) and a dummy Atto$_{488}$-n* (green). Figure 5c–e display the number fraction PC in prototissue (PC$_{prototissue}$) and free PC (PC$_{free}$), as well as the size of the prototissue as a function of five palindrome densities at the PC

surface by statistical image analysis after 1 h reaction time (see also Supplementary Fig. 11). The number of PCs assembled in prototissues, as well as the total number assembled within prototissues, increases with increasing the surface density of the palindromic sequence at the shell (Fig. 5d, e). Prototissue formation is completely absent at 0% surface density, but starts to appear already at 25% degree of functionalization; even though with relatively small aggregates of only ca. 6 PCs. Due to similar reaction time in the metabolic reaction cycle, we can estimate that at least 65% of strand displacement occurs in a metabolic reaction cycle (as presented in Fig. 5a, b), because a similar prototissue formation is obtained therein.

Since the statistical analysis clearly shows that already relatively small degrees of palindrome surface functionalization are sufficient to induce prototissue formation, and since not all of the produced Cy5-n*-p products produced in an active PC are needed for a complete functionalization of a shell, we hypothesized that an excess of those might be used as a chemical messenger to activate otherwise dormant PCs (without Dz in the core) for assembly in a sender/receiver setup. To realize such a sender/receiver setup, we mixed active, DZ-containing PCs (sender, green channel, labeled with Atto$_{488}$-n$_{short}$*) with dormant receiver PCs (blue channel, labeled with Atto$_{565}$-n$_{short}$*, no Dz in the core) and added a high concentration of Subs-4 (~7 equivalent to the Dz at the PC core). Indeed, first, the known rapid color change of the active PC shell (green to magenta) with subsequent prototissue formation of the sender PCs occurs (Fig. 5f, g, and Supplementary Fig. 12). Interestingly, after 4 min of Subs-4 addition, the receiver PCs, however, also start forming prototissue-like structures. The blue fluorescence of the shell decreases, while a slight magenta color appears at the receiver PC shell. Therefore, the shell turns blue to purple and pink in the merged CLSM channel. This observation confirms that some of the product (Cy5-n*-p) generated in the sender PCs is released and triggers the DSD at the catalytically inactive receiver PC shells substituting the original Atto$_{565}$-n$_{short}$* strands hence driving prototissue formation of dormant receiver PCs. Importantly, the sender PCs can be unambiguously localized because they have both a magenta core due to the accumulation of the magenta-fluorescent product from the Dz reaction and a magenta shell from the DSD at the PC surface, while the receiver prototissues have a shell labeled in blue and magenta (originating from inter-PC signal transport and DSD) but without a magenta core (Fig. 5g, after 900 s of Subs-4 addition; sender and receiver prototissue 1 and 2, respectively). Two specifically highlighted areas clearly show these differences for a prototissue formed from sender PCs and one formed from receiver PCs (Fig. 5g, i). We

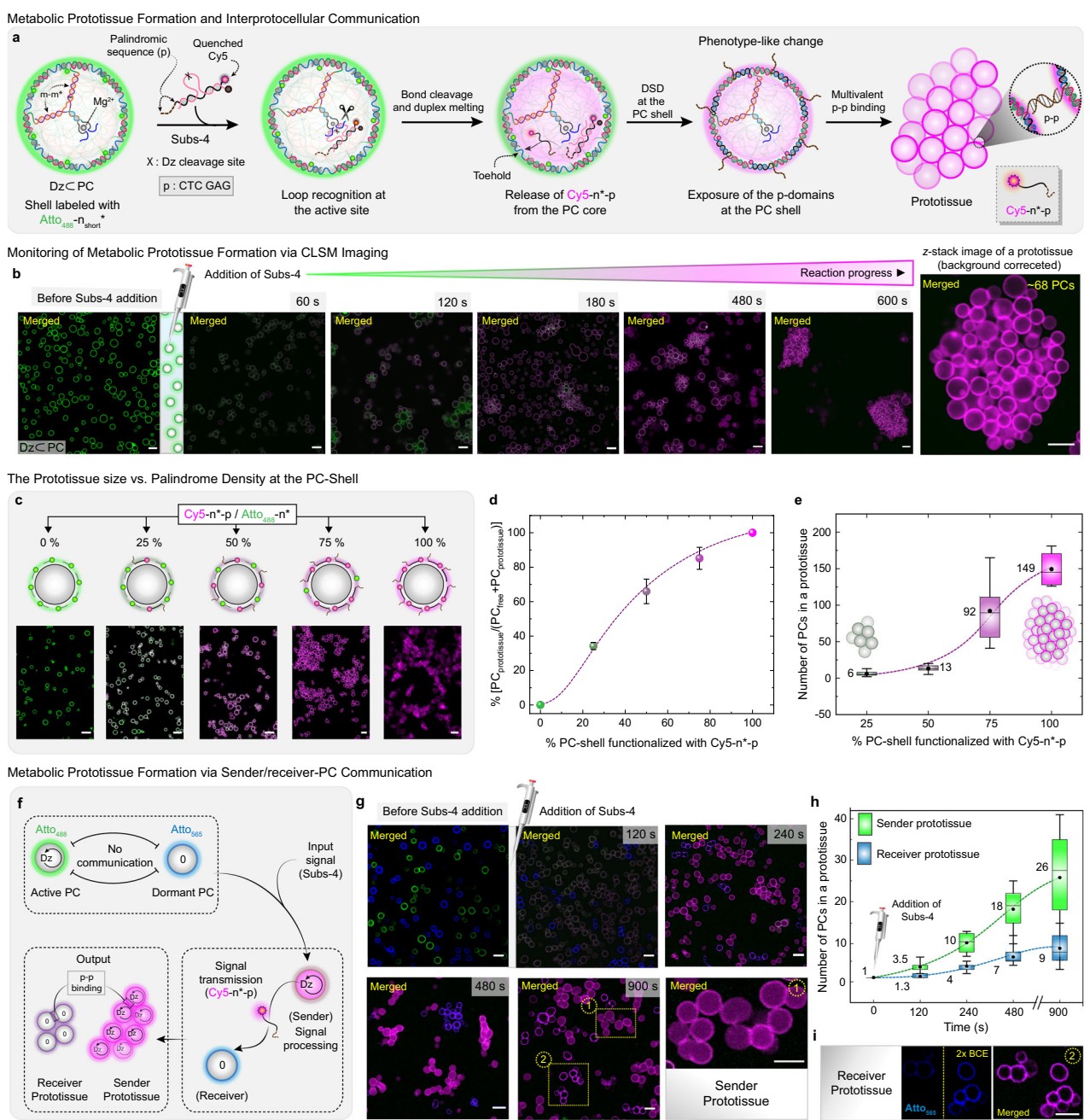

**Fig. 5 Metabolic bond cleavage and downstream prototissue formation via establishing interprotocellular communication. a** Schematic representation of a downstream cascade, DCBC → BCDM → DSD reaction at PC shell→ multivalent palindromic interprotocellular duplex formation, using Subs-4 as an input signal. **b** The time-dependent CLSM image of $Atto_{488}$-$n_{short}$*-labeled Dz⊂PCs before and after the addition of Subs-4. At $t = 10$ min, the CLSM image represents a prototissue (~68 PCs) built upon multivalent p-p* interactions. **c** Illustration and CLSM images of prototissues from five PC-mixtures, in which the palindromic-density at the PC shell is varied from 0 → 100%. Merged channels represent $Atto_{488}$ (green) and Cy5 (magenta) fluorescence. **d, e** statistical distribution of $PC_{prototissue}$ number fraction and size of the prototissue as a function of palindromic-density at PC shell. **f** Schematic illustration of downstream interprotocellular signal transduction and the growth of sender and receiver prototissues in a mixture containing active $Atto_{488}$-$n_{short}$*-labeled (green) PCs (Dz⊂PCs), and the dormant $Atto_{565}$-$n_{short}$*-labeled (blue) PCs. **g** Time-dependent CLSM images of interprotocellular communication and growth of prototissues in a mixed-PC system. A gradual increase of magenta fluorescence is observed at the shell of the active green PCs substituting the green fluorescence leading to the formation of sender prototissue. The shell color of dormant PCs changes to pink following the receiver prototissue formation within 4 min Subs-4 addition. The merged channel represents $Atto_{488}$ (green), $Atto_{565}$ (blue), and Cy5 (magenta) fluorescence. **h** Time-dependent growth of sender and receiver prototissues. **i** The zoomed-in images of two receiver prototissues (blue and merged channel). BCE represents brightness and contrast enhancement. The error bar in **d** represents the standard deviation of two different sample means at a particular palindromic-density ($N = 358$ PC-counts over two samples). For **e**, **h** the mean and the median line over the prototissue population are represented inside the box. The box represents the five-number summary of the $PC_{prototissue}$ date set and is extended from the first quartile to the third quartile. The whiskers represent the standard deviation of ten prototissue counts. The dotted lines are a guide to the eye. The scale bar = 5 μm. Condition: [Dz⊂PCs] = 5 μM, [Subs-4] = 35 μM in TE buffer at 35 °C.

also refer the reader to line segment analysis of the CLSM images of sender and receiver prototissue in Supplementary Fig. 13. Due to the spatial delay in signal transfer from the sender to the receiver PCs, the metabolic growth of prototissues has slower kinetics in the case of the receiver prototissue with respect to the sender prototissue. Interestingly, the delayed activation of the receiver PCs also leads to largely self-sorting structures with clear domains of sender PC prototissues that form more quickly and receiver PC prototissues that assemble preferentially with themselves later due to higher mobility of them as long as they are not assembled. Figure 5h summarizes the corresponding time-dependent analysis of the respective prototissue growths. These results confirm that rationally designed signal transduction cascades—combining catalytic Dz-based conversion, DSD reactions, and multivalent binding scenarios—allow for establishing a metabolic PC system, in which a signal gets processed into an effector to establish phenotype-like changes and communication. Moreover, the metabolic downstream signal transduction from active PCs (sender) to the dormant PCs (receiver), leading to the interprotocellular communication and morphological adaptation, resembles, albeit on a simplistic level, adaptation pathways in living cellular communities.

In summary, we introduced generic and versatile signal processing routines into communities of PCs by exploiting established DNAzyme and DSD reaction scenarios. The approach has the encapsulation of a DNAzyme inside the crowded macromolecular interior of a highly programmable all-DNA PC at its core. Gratifyingly, we found that DNAzymes have an enhanced activity inside such PCs, and they can be used to cleave RNA linkage-containing DNA signaling molecules to release output strands (metabolites). Those metabolites were shown to induce downstream processes by spatial relocation from the core to the shell and site-selective strand displacement reactions. The ensuing metabolic adaptation can be evolved to work on different levels of function and complexity. Simple phenotype-like changes in individual PCs can be evolved to induce attractive interactions between PCs of the same type by presenting multivalent binders, and even communication to a secondary PC population can be established. These metabolic features are reminiscent of the functional behavior of a living cell at simplistic levels.

Even though DNAzymes have been established for several purposes, we believe that our approach to using this DNAzyme in a confined state and establishing a DNA-based downstream reaction cascade based on using the catalytic conversion to lead to PC adaptation provides valuable insight into designing minimalistic life-like abiotic systems that can process, translate, communicate, and relay DNA-based signals in more complex sensory environments. Taking into account recent advances in sensing antibodies[55], as well as the widely developed fields of DNA aptamers[56] and DNAzymes[57], it appears feasible to achieve a broader sensing capability and to ultimately arrive at situations where communication between a natural cell and a PC can be established.

## Methods

**Synthesis of circular ssDNA template and multiblock DNA polymer (a general procedure for rolling circle amplification).** The synthesis of the ssDNA polymers was adapted and slightly modified from our previous report[42]. The stock solutions of 5′-phosphorylated templates and their corresponding ligation strand (see Supplementary Table 1 and Supplementary Fig. 1) were mixed (in equimolar stoichiometry) to attain a final concentration of 1 μM in TE buffer (Invitrogen; 10 mM Tris(hydroxymethyl)aminomethane pH = 8 and 1 mM EDTA) containing additionally 100 mM NaCl (total volume 100 μL). The concentrations of the strands are kept low to prevent inter-strand hybridizations. The buffered mixture was heated up to 85 °C (for 5 min) at a rate of 3 °C s$^{-1}$ and slowly cooled down to 20 °C at 0.01 °C s$^{-1}$. After annealing the strands, 20 μL of 10 X commercial ligase buffer (Lucigen; 500 mM TRIS-HCl, 100 mM MgCl$_2$, 50 mM dithiothreitol, and 10 mM ATP), 75 μL of water, and 5 μL of T$_4$ ligase (4 U μL$^{-1}$) were added in the reaction

tube containing 100 μL of the template strand, stirred (10 min, 400 rpm) and left to react for 4 h at 30 °C. The T$_4$ ligase was then denatured by heating the reaction mixture for 20 min at 70 °C. Then 10 μL of exonuclease I (Lucigen; 40 U μL$^{-1}$) and 10 μL of Exonuclease III (Lucigen; 200 U μL$^{-1}$) are added, and the mixture was kept overnight at 37 °C on a thermo-shaker (Eppendorf) with gentle stirring (300 rpm) to digest unreacted ligation strands and non-circularized templates in solution. Then the exonucleases were subsequently deactivated by heating the reaction mixture at 80 °C for 40 min. The circular templates were purified using the Amicon Ultra-centrifugal filters with a 10 kDa cut-off (Merck Millipore) and washed three times using TE buffer over the same filter. The concentration of circular templates was measured by the use of a ScanDrop (Jena Analytic) spectrophotometer, and the solutions were diluted to 1 μM using TE buffer. The template synthesis was repeated in multiple batches, and a stock solution of the circular template was prepared to avoid batch-to-batch discrepancy.

To synthesize multiblock ssDNA polymers, 10 μL circular template (1 μM) were mixed with 20 μL of commercial 10 X polymerase buffer (Lucigen; 500 mM TRIS-HCl, 100 mM (NH$_4$)$_2$SO$_4$, 40 mM Dithiothreitol, 100 mM MgCl$_2$), 2 μL of exonuclease resistant primer (10 μM in TE buffer), 5 μL of Φ$_{29}$ Polymerase (Lucigen; 10 U μL$^{-1}$), 20 μL of pyrophosphatase (New England Biolabs; 0.1 U μL$^{-1}$), and 10 μL of an adjusted dNTP mixture (total dNTP concentration of 5 mM, the percentage of each base correspond to the expected sequence composition in the ssDNA polymer), and 134 μL of ultrapure nuclease-free water. The reaction mixtures were kept at 30 °C for 60 h on a thermo-shaker with a gentle stirring (300 rpm). The solution containing ultralong ssDNA polymer was subjected to temperature-induced cleavage for 15 min at 95 °C, and the resulting solution was then washed through Amicon Ultra-centrifugal filters with a 30 kDa cut-off (Merck Millipore) (three to four times) using 350 μL of TE buffer. The amount and purity of the ssDNA polymers were determined using a ScanDrop (Jena Analytic) spectrophotometer. The pyrophosphatase enzyme was added to prevent the formation of insoluble magnesium pyrophosphate as a side product that hinders the polymerization process and is also accountable for resulting in so-called nanoflowers and might cause misperception during the microscopic investigations.

**Standard procedure for all-DNA protocells (PCs) preparation.** The synthesis of the PCs was adapted and slightly modified from our previous report[42]. The PCs were prepared by mixing the ultralong ssDNA-multiblock polymers to attain the target concentration (for p(A$_{20}$-m), typically, around 0.16 g L$^{-1}$, which corresponds to the repeating unit (r.u.) concentration of 10 μM and for p(T$_{20}$-n), around 0.03 g L$^{-1}$, which corresponds to [r.u.] = 1.67 μM) in TE buffer (pH 8). The stoichiometric ratio of p(A$_{20}$-m): p(T$_{20}$-n) = 6: 1 represents an optimized empirical condition in order to obtain well-defined PCs with minimized free p(T$_{20}$-n) amount in solution. The mixture was heated to 95 °C for 15 min (heating and cooling rate 3 °C s$^{-1}$) for homogenization and thermal cleavage of long ssDNA polymer chains (no phase-separation) using a thermocycler. A solution of MgCl$_2$ (2 M) was added to the mixture to attain the final concentration of 50 mM (for phase-separation of p(A$_{20}$-m) at a $T_{cp}$ of ~47 °C) and subsequently heated to 95 °C for 12 min with heating and cooling ramps of 3 °C min$^{-1}$. After the formation of the PCs, a stoichiometric amount of corresponding fluorescently-labeled barcode* oligomeric sequences (Atto$_{488}$-m*, Atto$_{565}$-n* or Atto$_{647}$-n*) were added to the solution and left the colloidal solution for 2-3 h at room temperature with a constant stirring (500 rpm) to hybridize with the PC-barcodes before monitoring them via CLSM imaging. The thermal reduction of the molecular weight of the RCA polymers is crucial to obtaining spherical PCs instead of unwanted agglomerations. Usually, the PCs are stable for several weeks, even though they slowly sediment at the bottom of the tube. The PCs can be sedimented and redispersed numerous times after sequential post-synthetic functionalization at the interior and the shell.

**Encapsulation of the DNAzyme in the PC interior.** A stock solution of annealed DNAzyme (Dz, composed of Split-Dz1 and Split-Dz2) (200 μM) was added to a freshly prepared PC medium (50 μL, [r.u.] or [barcode-m] = 10 μM) to functionalize the m-barcodes inside the PCs with Dz entirely. The DNAzyme-loaded protocells (Dz⊂PC) were left for 2 h at room temperature with continuous stirring (400 rpm). The Dz⊂PCs were centrifuged (12,000 rpm, ~8 min) and redispersed with TE buffer containing 50 mM MgCl$_2$. The purified DNAzyme-loaded PC solution was used for further experiments.

**Quantification of Dz-induced bond cleavage in the PC interior.** Dz⊂PC (10 μM encapsulated Dz) stock solution is diluted five times using a TE buffer solution with 10 mM NaCl on a 384-well plate and kept inside the Tecan plate reader for 5 min to equilibrate at the desired temperature. A stock solution of the substrate was injected into the PC solution (1 μL, 5 μM), and the fluorescence intensities were recorded (for Cy5: excitation 629 nm, emission 679 nm, bandwidth interval 10 nm, settle time 0.5 s) every 20 s interval over 1 h period. Before every measurement, the plate was subjected to an orbital shaking (200 rpm, 3 s). The concentration of the uncaged product (hence the % of bond cleavage) throughout the reaction was calculated from a calibration plot of the product Cy5-appended oligomer (at various concentrations). In the case of catalysis in solution, the Dz was hybridized

with an m-barcode sequence, and the final concentration was maintained at 2 μM in TE buffer containing 50 mM MgCl$_2$ and 10 mM NaCl before the substrate addition.

**CLSM monitoring of RCM-induced uncaging inside the PC interior**. The shells of the freshly prepared washed Dz⊂PC (10 μM encapsulated Dz) were hybridized with Atto$_{488}$-n*/ Atto$_{488}$-n$_{short}$*in order the visualize (excitation: 488-line, 364-well plate) the protocells before the addition of the substrates (Figs. 2h, 3a, 4h and 5b, 5g). A stock solution of the substrate (0.5 mM) was added to the reaction medium (30 μL) to a final concentration of 25–30 μM. The Dz⊂PCs were then visualized by excitation with two lasers: 488 nm (for shell) and 638 nm (for the Cy5 product). In some cases, the area of interest for the CLSM measuring was changed after capturing a few images to prevent the photobleaching of Cy5. During the reaction, the well plate was kept at 30 or 37 °C using a temperature-controlled microscopic stage. For the mixed-PC (active and dormant) experiment (Figs. 4h and 5g), three laser lines were used to excite the PCs ——488, 554, and 638 nm for Atto$_{488}$-n$_{short}$*, Atto$_{565}$-n$_{short}$*, and Cy5 product, respectively—and the images were recorded with minimum crosstalk amongst the detectors.

**Fluorescence recovery after photobleaching (FRAP) experiments**. The Dz⊂PCs (shell labeled with Atto$_{488}$-n*) were first imaged using a low intensity of the corresponding laser in a glass-bottom 364-well plate. A stock solution of Subs-1 (0.5 mM, 1.5 μL) was added to the PC solution, and the reaction was monitored over 20–30 min. The FRAP was done after the completion of the reaction. Photobleaching was attained using 100% intensity on both the 488 and 638 lines (≤0 mW in the focal plane). Seven steps of bleaching (each step 6 s) were performed on the Dz⊂PCs to deplete the fluorescence fully at the region of interest, and the 20 images were recorded every 10 s of the post-bleach sessions. The images were compared before and after the photobleaching steps. The data were presented in Fig. 3b, c.

**Flow cytometry measurements**. The data were collected on a Gallios flow cytometer (Beckman Coulter). Atto$_{488}$ and Cy5 were excited with a 488-nm or 638-nm laser and detected using a 525/40-nm or 660/20-nm bandpass filter, respectively. PCs were identified in the FSC/SSC scatter plot. PCs without Atto$_{488}$ and Cy5 fluorescence were excluded from the analysis (<3% of all PCs, Supplementary Figs. 5d, 8c, d). Approximately 500 PCs were measured per second. Acquired data before and after the addition of the substrate was concatenated and gated using FlowJo (v10.6.1, Becton, Dickinson, and Company). Binned medians were calculated over 4 s of acquisition time, and data were plotted using ggCyto in R 4.1.1. The concentration of encapsulated Dz in the PC is 0.5 μM for all the cytometry measurements.

**Image treatment and analysis**. The series of fluorescence images were processed using ImageJ (Fiji) by first applying a background subtraction. Then, the temporal profiles of both the magenta and green channels in the region of interest were extracted. The values plotted correspond to the magenta and green channels with the reaction time after normalizing with respect to the initial magenta and green intensities. The fluorescence intensities for the line segment analysis were presented without normalization.

**Statistics and reproducibility**. The reproducibility of each experiment was confirmed using three independent PC suspensions. No technical replicates were reported. For the fluorescence intensity measurements (Figs. 2c and 4c), three samples were prepared in three wells of a 386-well plate, and the experiments were run parallelly. For the microscopic analysis, corresponding changes over a large area with ~40–50 PCs were recorded and analyzed. The reproducibility of these experiments was checked with different PC batches and by imaging them using the same experimental parameters. Different areas were imaged in the kinetics to avoid the influence of photobleaching. For the statistical analysis, ~400 PCs from two samples were imaged and counted both in the free state as well as in the prototissue to prepare the box plots (Fig. 5d, e, h).

**Reporting summary**. Further information on research design is available in the Nature Research Reporting Summary linked to this article.

## Data availability

The source file also contains the numerical values of the number of protocells involved in flow cytometry kinetics, presented in Figs. 2f–g and 4d–e. Additional supporting data are available from the corresponding author upon request. Source data are provided with this paper.

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

## Acknowledgements

We would like to thank Oliver S. Thomas for help with flow cytometry data analysis and plotting. A.S. acknowledges the support of the Alexander von Humboldt Foundation. We acknowledge the support by the European Research Council Consolidator Grant to A.W. (M3ALI (agreement 677960)). A.W. acknowledges generous support from the Gutenberg Research College in the framework of a Gutenberg Research Professorship. M.H. and W.W. acknowledge support by the Deutsche Forschungsgemeinschaft (DFG, German Research Foundation) under Germany's Excellence Strategy CIBSS—EXC-2189—Project ID: 390939984.

## Author contributions

A.S. and A.W. conceived the project and designed the experiments. A.S. and M.H. performed the Cytometric experiments. A.S. performed the other experiments. W.L. optimized the palindromic sequences. A.S. analyzed the data and prepared the draft manuscript. A.W. and W.W. supervised the project. All authors commented on the manuscript.

## Funding

## Competing interests

The authors declare no competing interests.
