## [Peer Review File · Nature Communications]

Signal-Processing and Adaptive Prototissue Formation in Metabolic DNA ProtocellsReviewers' comments:

Reviewer #1 (Remarks to the Author):

Signal-Processing and Adaptive Prototissue Formation in Metabolic DNA Protocells"
NCOMMS-21-41694

1/ This is a well-written manuscript that demonstrates three logic-based downstream behaviours that arise from DNAzyme-activity in all-DNA coacervate-like protocells. The design is well thought out according to standard DNA nanotech logic, but the end processes are not so surprising or extensive. The studies are somewhat premature and the advances incremental; I therefore cannot recommend publication in NCOMM.

2/ Whilst I enjoyed reading about the work, it was not clear to me what the focus of the research was and how it advanced previous studies on protocellular systems. All-DNA protocells have been introduced previously by the authors (refs 36,37, 49) as well as others (Walczak et al., NCOMM 12, 4743 (2021) [not referenced]), so the underlying design principles and methodology concerning protocell construction appear to be well-established. Also, use of protocells as micro-reactors, operation of DSDs in protocells, and the proposed novelty of incorporating DNAzymes into protocell activity have recent precedents (for example, see Lui et al; ACIE 2020, 59, 6853 – 6859, which includes work on G4-hemin DNAzymes in coacervate droplet micro-reactors).

3/ In general, the sequence of data presented in the panels of each Figure tends to confirm a single central result rather than bringing new observations to the experiments. A similar methodological approach is essentially used in the three examples making the presentation a little repetitive. As a consequence, minimal additional knowledge is gained by using the different methods; in this regard, many of the results are confirmatory, which in other situations would place them in the SI.

4/ Several of the claims made by the authors are not justified. In particular, the title emphasizes "Adaptive Prototissue Formation" and this part of the work (example 3) is premature and hardly worth publishing at this stage. The authors demonstrate protocell aggregation, but no collective ("tissue") properties are described. Indeed, no quantitative analysis of aggregation behaviour – statistics, time-dependence etc – are included. Similarly, referring to the displacement of DNA in the shell region and a concomitant change in fluorescence as a change in "phenotype" is fanciful.

5/ Other comments:

a/ Fig 2c; is there any difference in the fluorescence output at longer time periods; from the plotted data it looks like the bulk system is slower but is increasing towards the threshold level seen in the PC system. Moreover, the threshold level seems to be decreasing (leakage into the bulk?).

b/ There appears to be a limit on DNAzyme activity due to the output being captured in the protocells. This suggests that use of the system as a micro-reactor is limited. The micro-reactor can be replenished by heating for 60 min, but this potentially compromises the actual use of the system. Moreover, as regeneration requires 60 min of external processing, it seems inappropriate to call the protocell system "metabolic" or "autonomous". I note that the term "metabolic DNA" is used in the title of the manuscript, and again this seems to be an unsubstantiated claim by the authors.

c/ Fig 3a,b – these results do not seem surprising as it has been already established in previous work that the protocells have a liquid-gel core-shell structure. They could be placed in the SI.

Reviewer #2 (Remarks to the Author):

Samanta et al. present an interesting route to establishing programmable behaviours in synthetic

micro-compartments (protocells). Using all-DNA capsules prepared by interfacial-arrested liquid-liquid phase separation, they exploit an internal catalytic activity to induce two main downstream processes, namely a change in the shell fluorescence (which is described as a “phenotype-like” modification) and DNA hybridization-mediated compartment aggregation (as a form of communication between different protocells) via dynamic strand displacement reactions.

This work is timely due to the increasing interest in assembling synthetic cells and communities capable of programmable, versatile, and collective behaviours. The choice and design of the experimental system to demonstrate these dynamic processes is elegant, and the smart use of DNA nanotechnology allows precise control over the system.

The manuscript is very clear and concise, experiments are well performed, and conclusions are nicely supported by the results. In particular, confocal images look great (I appreciate that the authors show many capsules in the field of view, as this makes the observations more robust), and well-thought FACS analyses provide good statistical information about the system. Perhaps my only minor reservation would be on the last part on prototissue formation that I feel lacks a bit of experimental investigations/statistical insights compared to the other parts (see detailed comments below). But other than that, and for all the above reasons, I strongly recommend publication in Nature Communications.

Minor comments:

1. The system requires relatively large molecules to be able to enter and exit the capsules (e.g. the DNAzyme, the cleaved oligonucleotides etc.). Do you have an idea of the typical molecular weight cut-off of the gelled shell?
2. On Fig. 2a, the scheme indicates “catalyst recovery”. This step requires heating to displace the cleaved product, could this be specified on the scheme for clarity?
3. How do the authors explain the higher activity of DNAzyme inside the capsules compared to bulk conditions (Fig. 2c)?
4. On Page 8, the authors write “In contrast, the red fluorescence at the PC core (ROI 3) exhibits almost complete recovery after each bleach cycle, confirming rapid diffusion of the product inside the PC core but for some remaining stickiness to the binding site”. This sentence is not clear.
5. On Figs. 3e,f: is there a washing step after incubation at 37°C? If not, why is red fluorescence not seen anymore in the samples (e.g. both outside and inside the capsules)? Or does the product become so dilute that it cannot be seen in the outer phase due to confocal imaging (thin slice of sample imaged)?
6. Additional details could be given on the prototissue formation part, e.g.:
 - are the samples continuously shaken during the experiment? If not, how can the protocells bind to each other (I guess they settle at the bottom of the well over time)?
 - why do some capsules seem to remain green in the prototissue?
 - can the authors quantify the surface density of binding oligonucleotides? Is a critical surface density required for a protocell to bind to another one?
 - what is the average number of neighbouring protocells in a prototissue?
 - could there be a way to control the final size of the prototissue (e.g. by changing the number density of protocells...)?
7. For consistency, times should be given in the same unit (minutes or seconds) in all experiments involving DNAzyme activities (see, e.g. Figs. 2c,f,g are in seconds, but Fig. 2h is in minutes; same for Figs. 4c,d,e in seconds but Fig. h in minutes...)

Reviewer #1 (Remarks to the Author):

1/ This is a well-written manuscript that demonstrates three logic-based downstream behaviours that arise from DNAzyme-activity in all-DNA coacervate-like protocells. The design is well thought out according to standard DNA nanotech logic, but the end processes are not so surprising or extensive. The studies are somewhat premature and the advances incremental; I therefore cannot recommend publication in NCOMM.

We thank the reviewer for his/her encouraging words. However, we do not believe that the studies are incremental or premature. Every step, from confined catalysis to release of a caged strand to downstream strand displacement to emerging interprotocellular palindromic interactions, needed specific design, optimization, and execution. We have developed the project step-by-step depicted in each Figure and finally developed prototissue formation using an in-situ confined catalytic reaction. Even though concepts may appear very logical in DNA nanoscience, the implementation of it in crowded environments and in multistep cascades is far from trivial regarding finding suitable experimental conditions, as well as preventing unwanted leakage of circuits.

In the revised work, we have now substantially extended the studies with respect to prototissue formation, and do not only provide significantly more quantitative insights, but also provide examples of communication and cross-activation of otherwise dormant protocells in sender/receiver settings.

2/ Whilst I enjoyed reading about the work, it was not clear to me what the focus of the research was and how it advanced previous studies on protocellular systems. All-DNA protocells have been introduced previously by the authors (refs 36,37, 49) as well as others (Walczak et al., NCOMM 12, 4743 (2021) [not referenced]), so the underlying design principles and methodology concerning protocell construction appear to be well-established.

Unfortunately, the reviewer has not grasped the key concept of this manuscript. Yes, we have developed/reported the all-DNA protocell model in our previous papers, but this work does not discuss the design or modification of the protocells. We have been using the DNA protocell model here as a platform to re-enact life-like behavior, including catalytic signal transformation in soft-hydrodynamic confinement as well as molecular communication scenarios, leading towards downstream functional/morphological change and signal-induced hierarchically transduced prototissue assembly. We are also not just encapsulating well-known multi-enzyme cascades to just show a catalytic reaction. Here, we establish a downstream reactivity pathway, which goes much beyond just making a reaction work in a protocellular entity.

Yes, there are reports of all-DNA microparticles {from Takinoue and Di Michele's group (citation added)} using the crosslinking strategy of the multivalent tiles. Again, however this article is not about making such particles, but using them for molecular systems engineering.

In an exaggeration, this study is not about a "chemically new compound (protocell)", but focuses on progress in system-level research, which is one of the biggest drivers in chemistry and artificial cell research.

Also, use of protocells as micro-reactors, operation of DSDs in protocells, and the proposed novelty of incorporating DNAzymes into protocell activity have recent precedents (for example, see Lui et al.; ACIE 2020, 59, 6853 – 6859, which includes work on G4-hemin DNAzymes in coacervate droplet micro-reactors).

We again believe that these papers are mentioned out of context. We are not encapsulating well-known enzyme-cascade (that also functions in free solution) in classical coacervates or in hydrogel to just study catalysis or make dyes. Our goal is to use catalytic transformation (encapsulated in crowded confinement using DNA-based bindings)

to generate a signal (DNA strand) that starts a downstream change in protocell functions and eventually makes the protocells communicate with each other leading to the prototissue formation (system level). To the best of our knowledge, coupling enzymatic reaction with DNA nanotechnology to realize functional and morphological changes in protocells and prototissue formation is still unprecedented. It is the system-level effect that is important, not that we use a specific type of catalysis.

This work solidifies our hypothesis of metabolic downstream signal propagation in crowded protocells (to mimic living systems) by combining several critical DNA-nanotechnology tools and orchestrating them in a synchronized fashion (Endogenous bond cleavage → strand release from the core of protocells → strand displacement at the shell → change of color of the shell → palindromic recognition → prototissue formation). We also believe the catalyst loading using orthogonal barcode functionalization is an important advantage of our protocell model because it provides spatial control rather than electrostatic capture of enzymatic centers in a classical coacervate matrix. To the best of our knowledge, this type of catalytic network with downstream functional change in fully DNA-made confinement has not been shown in the literature. Furthermore, the newly introduced sorting behavior of the emergent prototissue formation is an example of how sender-receiver kind of communication can be established as a result of a metabolic transformation (added in the new MS, Figure 5f–i).

3/ In general, the sequence of data presented in the panels of each Figure tends to confirm a single central result rather than bringing new observations to the experiments. A similar methodological approach is essentially used in the three examples making the presentation a little repetitive. As a consequence, minimal additional knowledge is gained by using the different methods; in this regard, many of the results are confirmatory, which in other situations would place them in the SI.

We used various techniques to establish the stepwise progression from (i) the differences in the catalytic activity in confined and free state, to (ii) the downstream signal relocation from the core to the shell and, finally, to (iii) the formation of prototissue. Now (revision) we have extended this even to (iv) the communication between sender and receiver protocells with consecutive signal-induced activation of protocells for prototissue formation.

We have used several complementary techniques (such as spectroscopy, cytometry, and microscopy) to quantify the processes. Spectrofluorometric (for investigating catalytic efficiency), cytometric (for statistical analysis), and microscopic (for real-time visualization) techniques are used for specific analysis of interest, and were used purposefully to design the sequences for the next step. Each method serves a purpose that is not targeting the same objective (efficiency, statistics, morphology). Reviewer two appreciates this in his/her comments.

4/ Several of the claims made by the authors are not justified. In particular, the title emphasizes “Adaptive Prototissue Formation” and this part of the work (example 3) is premature and hardly worth publishing at this stage. The authors demonstrate protocell aggregation, but no collective (“tissue”) properties are described. Indeed, no quantitative analysis of aggregation behaviour – statistics, time-dependence etc – are included. Similarly, referring to the displacement of DNA in the shell region and a concomitant change in fluorescence as a change in “phenotype” is fanciful.

*Now we have added a substantial amount of experimental details in the prototissue section (Figure 5), which include **i**) a complete CLSM monitoring of the enzymatic kinetics showing the downstream color change of the shell (green to red) and prototissue formation, **ii**) a communication pathway between the active sender and dormant receiver protocells leading to the formation of largely sorted prototissue distributions via interprotocellular signal transduction and **iii**) a detailed statistical analysis related to the palindrome surface density (at protocell surface) and the size of the prototissue (Figure 1 below). We believe that with this addition, we support our claim of*

metabolic downstream signal transduction (in both intraprotocellular and interprotocellular fashion) and catalytic prototissue formation, hence justifying the title of the paper. Now we have restructured the abstract to get our message across more clearly.

5/ Other comments:

a/ Fig 2c; is there any difference in the fluorescence output at longer time periods; from the plotted data it looks like the bulk system is slower but is increasing towards the threshold level seen in the PC system. Moreover, the threshold level seems to be decreasing (leakage into the bulk?).

We have repeated the DNAzyme-catalyzed bond-cleavage in bulk with freshly ordered substrate solutions, and Figure 2c has been updated. The slight decrease in the PC system after 15 min of substrate addition can be attributed to the PC sedimentation at the bottom of the well-plate. Added to MS.

b/ There appears to be a limit on DNAzyme activity due to the output being captured in the protocells. This suggests that use of the system as a micro-reactor is limited. The micro-reactor can be replenished by heating for 60 min, but this potentially compromises the actual use of the system. Moreover, as regeneration requires 60 min of external processing, it seems inappropriate to call the protocell system “metabolic” or “autonomous”. I note that the term “metabolic DNA” is used in the title of the manuscript, and again this seems to be an unsubstantiated claim by the authors.

The reaction temperature was not changed during the experiment (Figure 3e, f, i) to force the product release. The temperature of the protocells was always maintained at ~37 °C throughout the experiment. As the cleaved product (for Subs-1) has a binding affinity (6 nts) at the DNAzyme active site, the product’s release was slow, which is not the case for the loop substrate (Subs-3 and Subs-4). The loop substrates are cleaved and diffused to the shell instantaneously, supported by cytometry and CLSM (Figure 4 and 5).

Now we included a time-dependent CLSM experiment depicting the in-situ product formation (for Subs-1) and release in a single protocell over an hour time window (Supplementary Fig. S7).

Supplementary Figure S7: Dz-catalyzed intraprotocellular product formation and the product release. a) Time-dependent CLSM images representing the bond cleavage of Subs-1 by the encapsulated Dz in the PC core. The shell of Dz < PC is labeled with Atto488-n* (green), and the evolving red fluorescence (Cy5) ensures the product formation. b) Time-dependent CLSM images depict the product diffusion from the PC-core to the surroundings,

leading to the decrease of the red color at the core. c) The overall change in Cy5 fluorescence, measured at ROI-1, during the catalytic product formation and release.

c/ Fig 3a,b – these results do not seem surprising as it has been already established in previous work that the protocells have a liquid-gel core-shell structure. They could be placed in the SI.

Previously, we showed the core dynamics of the pristine protocells. Here the fluorescence comes from the cleaved product. We have reduced the size of figure 3. However, we believe it is crucial to ensure the product diffusivity inside the protocell core, which is instrumental for transporting the cleaved substrate from the core to the shell. Due to some remaining interaction between the product and the DNAzyme this behavior cannot be taken for granted in a crowded environment with higher ionic strength than in the surrounding medium.

Reviewer #2 (Remarks to the Author):

Samanta et al. present an interesting route to establishing programmable behaviours in synthetic micro-compartments (protocells). Using all-DNA capsules prepared by interfacial-arrested liquid-liquid phase separation, they exploit an internal catalytic activity to induce two main downstream processes, namely a change in the shell fluorescence (which is described as a “phenotype-like” modification) and DNA hybridization-mediated compartment aggregation (as a form of communication between different protocells) via dynamic strand displacement reactions.

This work is timely due to the increasing interest in assembling synthetic cells and communities capable of programmable, versatile, and collective behaviours. The choice and design of the experimental system to demonstrate these dynamic processes are elegant, and the smart use of DNA nanotechnology allows precise control over the system.

We thank the reviewer for his/her supportive and encouraging comment.

The manuscript is very clear and concise, experiments are well performed, and conclusions are nicely supported by the results. In particular, confocal images look great (I appreciate that the authors show many capsules in the field of view, as this makes the observations more robust), and well-thought FACS analyses provide good statistical information about the system. Perhaps my only minor reservation would be on the last part on prototissue formation that I feel lacks a bit of experimental investigations/statistical insights compared to the other parts (see detailed comments below). But other than that, and for all the above reasons, I strongly recommend publication in *Nature Communications*.

We thank the reviewer again for the supportive words and mentioning the importance of this very work. We have now added detailed analysis of the prototissue part in the manuscript (in Figure 5), including additional prototissue kinetics, a communication-based self-sorting of the active and dormant prototissues, and detailed statistical analysis on the size of the prototissue (see below).

Minor comments:

1. The system requires relatively large molecules to be able to enter and exit the capsules (e.g. the DNAzyme, the cleaved oligonucleotides etc.). Do you have an idea of the typical molecular weight cut-off the the gelled shell?

Even though the shell is hydrogel-like in the FRAP experiments (Figure 3a), it is porous enough to easily let the DNA oligos and DNAzymes pass through. The core barcodes (m) can be entirely functionalized with DNAzyme since they

contain complementary barcode recognition domains (m^). We have also shown in a previous paper (Nat. Nanotechnol. 2020, 15, 914-921) that proteins like streptavidin ($\sim 4.2 \times 4.2 \times 5.6 \text{ nm}^3$) can also be transported from outside. In a recent study (manuscript in preparation), we realized that a polymer-barcode conjugate up to 60k molecular weight could go inside the protocell. We do not have an exact molecular weight cut-off but believe that this is not an essential part of the study, as comparably large (StAv) are known to diffuse through the shell.*

2. On Fig. 2a, the scheme indicates “catalyst recovery”. This step requires heating to displace the cleaved product, could this be specified on the scheme for clarity?

We thank the reviewer for addressing the confusion regarding the catalyst recovery experiment. The reaction temperature was not elevated during the course of the experiment (Figure 3e, f, i). The Subs-1 was added to protocell suspension, kept at 37 °C. We waited 60 min to show that more than 85% of the product could be released from the protocell core, and Subs-2 could be cleaved successively. We have now mentioned this clearly in the manuscript and also in the scheme. Moreover, we added a time-dependent CLSM experiment depicting the in-situ product formation and release in a single protocell over one hour (Supplementary Fig. S7).

3. How do the authors explain the higher activity of DNAzyme inside the capsules compared to bulk conditions (Fig. 2c)?

We believe the higher activity of the DNAzyme can be addressed by the “crowding effect” of the liquid DNA core, which we have also seen in our previous studies. Moreover, the different environmental conditions that DNAzyme experiences inside the PC-core compared to them in solution. The polarity of the liquid DNA environment and high ionic strength could play an important role.

4. On Page 8, the authors write “In contrast, the red fluorescence at the PC core (ROI 3) exhibits almost complete recovery after each bleach cycle, confirming rapid diffusion of the product inside the PC core but for some remaining stickiness to the binding site”. This sentence is not clear.

The sentence has been restructured.

5. On Figs. 3e,f: is there a washing step after incubation at 37°C? If not, why is red fluorescence not seen anymore in the samples (e.g., both outside and inside the capsules)? Or does the product become so dilute that it cannot be seen in the outer phase due to confocal imaging (thin slice of sample imaged)?

Yes, the product is diluted to the outside of the protocells. In this case, an XY plane was imaged ($\sim 5 \mu\text{m}$ above the glass surface) to obtain a better contrast with the background.

6. Additional details could be given on the prototissue formation part, e.g.: - are the samples continuously shaken during the experiment? If not, how can the protocells bind to each other (I guess they settle at the bottom of the well over time)? - why do some capsules seem to remain green in the prototissue? - can the authors quantify the surface density of binding oligonucleotides? Is a critical surface density required for a protocell to bind to another one? - what is the average number of neighbouring protocells in a prototissue? - could there be a way to control the final size of the prototissue (e.g., by changing the number density of protocells...)?

As requested by the reviewer, we have added additional experiments on the prototissue section of the paper (Figure 5b-g). The experiments developing metabolic prototissue were done in a 384 well-plate under the confocal microscope on a temperature-controlled stage without additional shaking. Firstly, the DNAzyme-loaded protocells (green shell) were imaged before the addition of Subs-4 (at 35 °C). The protocells were not chemically fixed on the glass bottom, so the Brownian motion is present. The Subs-4 solution was injected and mixed adequately in the

protocell suspension. As the DNAzyme-catalyzed cleavage reaction is fast, the palindrome (released from the core)-decorated protocells bind with each other in a kinetically driven process. The images are recorded near the glass surface. We also included a z-stack image (Figure 5b, 35 slices of XY planes) of the final prototissue structure.

The green protocells in the tissue represent incomplete substitution (DSD) of the green dye by the red DNA product. We have now optimized the substrate concentration to form a uniform prototissue (all red).

As we mentioned in the manuscript, the palindrome sequence has a low melting temperature ($T_m = 19^\circ\text{C}$), so it can only crosslink protocell shells in a multivalent fashion. We have now included an experiment in which the surface coverage of the protocells by the Cy5-appended palindrome sequence was varied from 0% to 100% (Figure 5c in the manuscript; Figure 1a, b below). We also plotted the number fraction of protocells in the tissues against the palindrome surface density (Figure 5d), indicating the decrease of free protocells in solution with increasing the crosslinker density.

Furthermore, a statistical distribution of the average number of protocells in prototissue has been correlated with the palindrome surface density (Figure 5e in the manuscript; Figure 1c below). These results represent that the size of the prototissue can be controlled by the concentration of the released palindromic product.

Figure 1. a) The CLSM images of the coexistence of free protocells and prototissue with varying the palindrome density at the protocell surface. b) The number fraction of protocells in the prototissue with different palindrome densities. c) The average number of protocells in the prototissue.

Moreover, beyond the requested data, we have now included a DNAzyme-catalyzed inter-protocellular communication pathway leading to the formation of “sender” and “receiver” –prototissues. See new Figure 5f–h.

7. For consistency, times should be given in the same unit (minutes or seconds) in all experiments involving DNAzyme activities (see, e.g., Figs. 2c,f,g are in seconds, but Fig. 2h is in minutes; same for Figs. 4c,d,e in seconds but Fig. h in minutes...)

All the labels have been changed to seconds.

REVIEWERS' COMMENTS

Reviewer #2 (Remarks to the Author):

The authors have addressed all my comments in this revised version. They have done changes to the main text and have included additional experiments that I think strengthen the significance of the work. Of particular relevance to my concerns, the experimental section on prototissue formation has been expanded with more detailed analyses on the size of prototissues, and additional experiments to demonstrate a communication-based self-sorting of active and dormant prototissues.

Based on these observations, I support publication in Nature Communications.

Reviewer #3 (Remarks to the Author):

I have reviewed the manuscript, particularly in the context of comments from Reviewer 1. I must take the authors' side here, as I think this work is hardly incremental and represents a very good addition to literature on DNA-based protocells. The authors have now cited related works from Takinoue and Di Michele groups, which is good (I may also suggest https://onlinelibrary.wiley.com/doi/full/10.1002/adfm.202202322#.YoOVVREKR_A.twitter and <https://www.biorxiv.org/content/10.1101/2022.03.24.485404v1>). However, the fact that the authors themselves, and others, have already introduced the concept of DNA-based protocell is not a deal breaker. The data in this manuscript look solid and some of the approaches innovative, so I would recommend publication.

The only point where I agree with Reviewer 1 is that framing colour changes as phenotypical changes is a little over the top, so I would recommend softening the authors' stance on this point.

In the context of prototissue formation, I would also add some references to the extensive literature on DNA-mediated adhesion and tissue formation in liposome based "protocells" (e.g. <https://www.nature.com/articles/ncomms6948>, <https://pubs.acs.org/doi/10.1021/acsnano.5b07201>, <https://pubs.acs.org/doi/10.1021/jp075792z>)

Reviewer #2 (Remarks to the Author):

The authors have addressed all my comments in this revised version. They have done changes to the main text and have included additional experiments that I think strengthen the significance of the work. Of particular relevance to my concerns, the experimental section on prototissue formation has been expanded with more detailed analyses on the size of prototissues, and additional experiments to demonstrate a communication-based self-sorting of active and dormant prototissues.

Based on these observations, I support publication in Nature Communications.

We thank the reviewer for his/her encouraging words and support.

Reviewer #3 (Remarks to the Author):

I have reviewed the manuscript, particularly in the context of comments from Reviewer 1. I must take the authors' side here, as I think this work is hardly incremental and represents a very good addition to literature on DNA-based protocells. The authors have now cited related works from Takinoue and Di Michele groups, which is good (I may also suggest https://onlinelibrary.wiley.com/doi/full/10.1002/adfm.202202322#.YoOVVREKR_A.twitter and <https://www.biorxiv.org/content/10.1101/2022.03.24.485404v1>). However, the fact that the authors themselves, and others, have already introduced the concept of DNA-based protocell is not a deal breaker. The data in this manuscript look solid and some of the approaches innovative, so I would recommend publication.

We thank the reviewer for his/her support and encouragement. We have added one more reference regarding the functions of all-DNA colloids (Ref 33). We thank the reviewer for the suggestion.

The only point where I agree with Reviewer 1 is that framing colour changes as phenotypical changes is a little over the top, so I would recommend softening the authors' stance on this point.

We thank the reviewer for the suggestion. In fact, the word "phenotype change" is used to stress the point that the active protocells can sense and process the environmental signal, and using a downstream pathway, they can differentiate themselves from the dormant cells (protocells with no active enzyme) by changing the functionality, as it happens in case of stem cells (also known as metaplasia, ref: <https://www.nature.com/articles/nrm761>). People have used the term "phenotype change" before in the protocellular context (Ref: <https://www.nature.com/articles/s41467-017-01161-8>, <https://pubs.acs.org/doi/10.1021/sb300125z>). We have changed the term to "phenotype-like change" in the manuscript.

In the context of prototissue formation, I would also add some references to the extensive literature on

DNA-mediated adhesion and tissue formation in liposome based “protocells” (e.g. <https://www.nature.com/articles/ncomms6948>, <https://pubs.acs.org/doi/10.1021/acsnano.5b07201>, <https://pubs.acs.org/doi/10.1021/jp075792z>)

We thank reviewer for the suggestions. We have added two more citations regarding the prototissue formation (Ref 53, 54).